# DG-MRM: Dynamic Graph and Multi-view Medication Information for Recommending Medication Combination

Guanlin Liu[1]
[1]Shandong Normal University
Jinan, China
liuguanlin0818@163.com

Zihao Liu[1]
[1]Shandong Normal University
Jinan, China

Xiaomei Yu[1,2,*]
[1]Shandong Normal University
Jinan, China
yxm0708@126.com

Xue Li[1]
[1]Shandong Normal University
Jinan, China

Xiangwei Zheng[1,2]
[2]China State Key Laboratory of
High-end Server & Storage
Technology
Jinan, China

Xingxu Fan[1]
[1]Shandong Normal University
Jinan, China

## ABSTRACT

An accurate and safe medication recommendation system play a vital role to assist medical practitioners in making prescriptions and enhance patient treatment outcomes. Existing approaches often treat electronic health record (EHR) data as sequences in training, failing to capture the complex dependencies between different medical events. Additionally, these systems encounter challenges in mitigating drug-drug interactions (DDIs). In response to these limitations, we propose a novel medication recommendation model named DG-MRM, which constructs dynamic graphs with multi-source EHR data and utilizes multi-view medication information from external knowledge database. Specifically, we leverage dynamic graph neural networks to investigate the temporal and structural relationships between different treatment data in patients' medical history, aiming to generate comprehensive patient representations with several longitudinal visits. Moreover, we capture internal-view medication molecule structure and functional-view interactions between medication molecules to generate safe medication combinations. Finally, our proposed DG-MRM is extensively evaluated on a benchmark dataset and the results reveal superior performance in terms of efficacy and safety. Comparative analysis with state-of-the-art drug recommendation models, DG-MRM significantly improves the accuracy of drug recommendations while maintains a low risk of DDIs.

## KEYWORDS

Medication Recommendation, Electronic Health Record, Drug-Drug Interaction, Dynamic Graph

## 1 INTRODUCTION

The advent of personalized medicine and the increasing complexity of patient data necessitate advanced decision-support tools in healthcare[2, 9, 17]. In particular, medication recommendation systems [5, 11] play a prominent part in enhancing treatment outcomes and ensuring patient safety by providing accurate medication suggestions. In recent years, the integration of Electronic Health Records (EHRs) into clinical decision-making processes has burgeoned, offering vast amounts of data that encapsulate detailed patient histories and treatment trajectories[6]. Accordingly, the

massive accumulation of EHRs have promoted the booming of intelligent medicine, such as medication recommendation (MR).

Despite advances in current approaches to medication recommendation, the potential of EHRs in crafting personalized treatment plans is yet to be fully exploited. Taking the patient representation as an example, which is an essential component in the MR tasks. While recent models like GAMENet[12] have begun incorporating patient's historical visit records to capture dynamic changes over time with Recurrent Neural Networks (RNNs), these models generally ignore the complex interactions among patients' historical visit sequences. Furthermore, these existing methods fail to capture the intricate dependencies between different medical events in a patient's medical history. Nevertheless, the patient's health history are crucial for developing a comprehensive understanding of the patient's health. In the other fundamental component in MR tasks, managing drug-drug interactions (DDIs) remains a significant challenge to generate safe medication combinations, which is critical for ensuring patient safety and the efficacy of treatment plans. For instance, models like GAMENet focus primarily on the interactions between drug molecules, whereas systems like SafeDrug[18] concentrate on modeling the structural properties of drugs. To date, no existing model in the field of medication recommendation effectively integrates both perspectives—intermolecular interactions and molecular structures—in a unified approach to model drugs thoroughly. This limitation highlights a crucial gap in the current medication recommendation systems, underscoring the need for a more comprehensive model to fully capture the complexities of DDIs.

To address the identified challenges in current drug recommendation practices, we propose a novel model named DG-MRM, which innovatively integrates dynamic graph-based techniques to enhance the patient representations and combines multi-view medication molecular information to reduce adverse DDIs. Specifically, in the patient representation module, RNNs are leveraged to capture the temporality in dynamic graphs, which incorporates both temporal and topological information from the patients' historical EHR data; and a multi-view medication information module is employed which combines medication molecular structure information with drug-drug interaction data to generate integral medication representations for accurate and safe medication recommendation. The main contributions of the paper are summarized as follows:

---
* Corresponding author.

- **Dynamic Graph-Based Modeling on EHR Data:** We model longitudinal patient records with constructed dynamic graphs that capture both the structural relationships and temporal dependence among medical events. Moreover, RNNs are leveraged to effectively learn the evolution of patient states over time, which enhances the accuracy and interpretation on EHR data.

- **Combine Multi-view to capture DDI information:** We incorporate the drugs into their molecular structures and sub-structures to capture critical DDI information. By integrating diverse drug knowledge into the medication representation, the DG-MRM model gains a deeper understanding of how drugs interact at the molecular and systemic levels, thereby enhances the safety and efficacy in medication recommendation.

- **Empirical Validation and Benchmarking:** Following the methodologies in recent studies, DG-MRM is rigorously evaluated on a free available dataset derived from the MIMIC-III database[6], which is a standard benchmark in the field of medication recommendation. The experimental results demonstrate substantial improvements over existing state-of-the-art methods with a faster convergence rate, specifically on the metrics of DDI rates, Jaccard similarity and F1 measure.

## 2 RELATED WORK

The prevailing studies in medication recommendation are mainly categorized into two types: instance-based approaches and longitudinal ones.

**Instance-Based Approaches:** Instance-based approaches focus on real-time patient medical data, which analyze the patients' current symptoms and medical records for treatment suggestion. For instance, LEAP[20] employs sequential decision-making and utilizes a cyclic decoder to model label dependence. Following this line of research, PPC[14] integrates patient demographics, diagnosis, and medication data with a tri-linear method to improve the accuracy in drug recommendation tasks. Despite the effectiveness gained in handling real-time data, these instance-based methods generally overlook the patients' longitudinal visit history, leading to reduced predictive performance on complex chronic conditions and thus limiting the comprehensiveness of treatment outcomes.

**Longitudinal Approaches:** Longitudinal approaches combine the historical sequence of a patient's medical visits to inform treatment recommendations. For instance, RETAIN[3] utilizes a two-level neural network with attention mechanism that not only processes sequential EHR data but also provides interpretability by highlighting influential past medical events in decision-making. DMNC[8] is equipped with a dual memory neural network, which enhances its ability to capture long-term dependencies in medical histories for accurate prediction. However, the above methods overlook the adverse impacts of DDIs. GAMENet[12] learns both adverse DDI knowledge and historical medication prescriptions with dynamic memory and graph convolutional networks(GCN) to tailor medication recommendations to individual patient profiles. SafeDrug[18] employs a dual-molecule encoder to capture both global and local molecular patterns in drugs. Moreover, SafeDrug

achieves safety in drug recommendation benefiting from a flexible loss function designed to manage DDIs in a controllable manner. Following this research direction, COGNet[16] utilizes an encoder-decoder architecture and a "copy or predict" mechanism to generate sets of medications, and enables personalized drug recommendations based on sequential EHR data. AKA-SafeMed[19] combines the representation of both patients and drugs with a multi-label classifier, a list of drugs is predicted for recommendation.

While existing longitudinal approaches have achieved initial success, they still exhibit significant limitations. First of all, these methods typically reduce complex medical histories to simple sequential data, failing to utilize the intricate structural relationships in different medical events. Moreover, current methodologies address adverse DDIs inadequately by focusing solely on intra-molecular structures within drugs or merely considering interactions between drug pairs. To address the limitations in existing medication recommendations, we propose a novel DG-MRM model that not only learns the temporal sequences in the evolution of patient medical records, but also integrates the structural interactions within different medical events in each visit. Furthermore, inspired by the MIRACLE framework[15], we innovatively considers both the internal-view medication molecule structure and the functional-view interactions between medication molecules that significantly enhances the precision and safety of medication recommendation.

## 3 METHOD

### 3.1 The DG-MRM Model

To tackle the challenges results from inadequate learning on patient and medication representations for medication recommendation, we propose a novel DG-MRM model, which constructs dynamic graph from patient's historical EHR data and integrates multi-perspective medication information, such as the internal-view medication molecule structure and the functional-view interactions between drugs, to enhance the safety and efficacy of medication recommendations. The DG-MRM includes four components: dynamic graph construction, longitudinal patient representation, multi-view medication representation and medication recommendation. Due to space constraints, we enclose the model frame graph in the appendix.

### 3.2 Dynamic Graph Construction

The EHR data are typically expressed as medical codes in sequences, while no explicit connections available among the different sequences of diagnoses, procedures and medications. To investigate the relationships of diverse medical codes in EHRs, we explore a graph structure for EHR data based on the methodology in GCT[4].

With original EHRs input, the mathematical statistics is leveraged to learn the implicit structural information $\hat{A}$ in EHR data for graph construction. Specifically, conditional probabilities between EHR features are utilized to guide the extraction of implicit structural information. In the medical applications, several practical scenarios are considered in graph construction. Firstly, as to the nodes with certain connection, a value of zero is assigned to the mask of M in the graph constructed. Additively, certain nodes of a patient's visits may not be connected currently, such as some diagnostic nodes do connect to the medication ones while have no connection to the other diagnostic or procedure nodes. In such a

circumstance, a mask M is added with the value of negative infinity explicitly during the attention generation process. Moreover, the self-connection of each node is essential, and they are assigned with positive infinity. Therefore, the mask M is defined as follows:

$$M_{jk} = \begin{cases} +\infty & \text{, if a connection of } n_j \text{ and } n_k \text{ is guaranteed} \\ 0 & \text{, if a connection of } n_j \text{ and } n_k \text{ is allowed} \\ -\infty & \text{, if a connection of } n_j \text{ and } n_k \text{ is not allowed} \end{cases} \quad (1)$$

With the mask settled, we use conditional probabilities to guide the attention calculation based on the diagnosis codes $d_i$, procedure codes $p_i$, and medication codes $m_i$ in the patients' visit records, which is defined as follows:

$$P(j|k) = \frac{n_{jk}}{n_j} \quad (2)$$

where $P(j|k)$ represents the appearance probability of node $k$ given the known node $j$. Here, $n_{jk}$ is the co-occurrence number of medical node $j$ and node $k$, and $n_j$ is the number of node occurred.

With the conditional probabilities $P(m_i|d_i)$, $P(d_i|m_i)$, $P(p_i|m_i)$, and $P(m_i|p_i)$ calculated, we develop a guidance matrix $P$ with a normalization operation to ensure that the values in each row are summed to one. Since the same dimension is gained in mask $M$ as that in the conditional probability $P$, the implicit structural information $\hat{A}$ is calculated as follows:

$$\hat{A} = \sigma(P + M) \quad (3)$$

where $\sigma$ is the calculation function.

With the implicit structure information $\hat{A}$ achieved, we develop the dynamic graphs structure with the patients' historical EHR data.

## 3.3 Longitudinal Patient Representation

*3.3.1 Visit Embeddings.* To extract the comprehensive medical information, we develop three embedding tables $E_d \in \mathbb{R}^{|D| \times \dim}$, $E_p \in \mathbb{R}^{|P| \times \dim}$ and $E_m \in \mathbb{R}^{|M| \times \dim}$ for diagnosis, procedure, and medication codes, respectively. Here, each row represents an embedding vector, and dim is the embedding dimension.

Given the set of diagnosis codes $D_t \in R^{|D_t|}$, procedure codes $P_t \in R^{|P_t|}$ and medication codes $M_t \in R^{|M_t|}$ for $t$-th visit of patient, combined with the embedding table, we can respectively obtain the embedding set of the diagnostic nodes $ED_t \in R^{|D_t| \times \dim}$, procedure nodes $EP_t \in R^{|P_t| \times \dim}$ and medication nodes $EM_t \in R^{|M_t| \times \dim}$ visited by the patient at the $t$-th visit. By connecting the above embedding sets, we can obtain the node embedding matrix $N_t \in R^{n \times \dim}$ in the subgraph of the patient's $t$-th visit, where $n = |D_t| + |P_t| + |M_t|$ is the total number of nodes in the subgraph.

*3.3.2 Parameter learning in dynamic graphs.* In the EHRs, a patient's historical visits are different in timestamps, which is crucral to investigate the relationships of diverse medical events and the evolution of the diseases in a patient. However, the traditional approaches are no longer applicable which combine graph neural networks with cyclic architecturesto generate the fixed node embeddings. Therefore, the dynamicity and temporality in EHRs pose significant challenges in learning patient representations with the patients visit data. To address these challenges, we construct

GCNs[7] indexed by time for each visit, and leverage the Gated Recurrent Units (GRU) [1] architecture to learn the dynamically updated hidden states in the dynamic graphs in the chronological order.

First at all, the EHR data of a patient's visit to the hospital is deployed on a snapshot subgraph in the dynamic graph structure. And then, a multi-layered GCN is deployed for each visit to learn the structural relevance among diverse medical events within a snapshot subgraph. Subsequently, Inspired by EvolveGCN[10], we employ Gated Recurrent Units (GRU) to dynamically update the parameters in GCNs, which is shown in equation 4:

$$W_t^{(l)} = \text{GRU}(H_t^{(l)}, W_{t-1}^{(l)}) \quad (4)$$

where $H$ is a medical event of a patient's visit input.

With the snapshots of subgraphs denoting the patients' visits, the core of EvolveGCN lies in its weight updating of GCNs with the evolution of patients' visits. In this way, we capture the structural relationships of diverse medical events in a visit of each patient. Continually, given the input data of an implicit structural information matrix $\hat{A}_t \in \mathbb{R}^{n \times n}$ and an explicit node embedding matrix $N_t \in \mathbb{R}^{n \times dim}$, the evolution of medical event nodes in the dynamic graph is formalized as follows:

$$H_t^{(l+1)} = \text{GCONV}\left(\hat{A}_t, H_t^{(l)}, W_t^{(l)}\right) \quad (5)$$

$$\tilde{D} = \text{diag}\left(\sum_j \hat{A}_{ij}\right) \quad (6)$$

where $l$ is the layer index of GCN, $W$ is the weight matrix, $\sigma$ is the activation function. Here, at the time step $t$, the initial embedding matrix is the node feature matrix input, i.e., $H_t^0 = N_t$.

*3.3.3 Patient Representations.* We use the evolved GCN to capture semantic information about each medical event from a patient's past $t-1$ visit and update the feature vector set of each medical event as follow:

$$\begin{aligned} &\{HD_{1:(t-1)}, HP_{1:(t-1)}, HM_{1:(t-1)}\} \\ &= \text{EGCN}(\hat{A}_{1:(t-1)}, ED_{1:(t-1)}, EP_{1:(t-1)}, EM_{1:(t-1)}) \end{aligned} \quad (7)$$

With the graph embeddings of both nodes and edges gained, the multi-head attention mechanism is leveraged to project the contextual embeddings into their integral embedding space.

$$hd_t = Attention(HD_t, HD_t, HD_t) \quad (8)$$

$$hp_t = Attention(HP_t, HP_t, HP_t) \quad (9)$$

where $t \in (1, T)$ and the feature set of $T$-th visit is the initial embedding set, i.e., $HD_T = ED_T$ and $HP_T = EP_T$. Here, the contextual embeddings $h_{d_i}$, $h_{p_i}$ and $h_{m_i}$ are generated for the diagnosis, procedure and medication codes respectively, each of which incorporate both the initial featural embeddings of nodes and the implicit structural information of edges from the patients' EHR dynamic graph. Subsequently, the GRUs to obtain the hidden diagnosis and procedure vectors $d_h^{(t)}$ and $p_h^{(t)}$ are formalized as follows:

$$hd_t' = GRU_d(hd_1, hd_2, \cdots, hd_t) \quad (10)$$

$$hp_t' = GRU_p(hp_1, hp_2, \cdots, hp_t) \quad (11)$$

Guanlin Liu[1], Zihao Liu[1], Xiaomei Yu[1,2,*], Xue Li[1], Xiangwei Zheng[1,2], and Xingxu Fan[1]

Combining the diagnosis embeddings with procedure ones, the longitudinal patient representation is generated by concatenating the embeddings with a feedforward neural network, which is defined as follows:

$$q_t = f([hd_t^{'}, dp_t^{'}]) \tag{12}$$

where $f(\cdot)$ is the transform function implemented with a single hidden layer fully connected neural network.

## 3.4 Multi-view Medication Representation

*3.4.1 Internal-view of Drug Molecules.* Inspired by the SafeDrug, we employ a message-passing neural network (MPNN) to extract drug atomic information from the drug molecular structure graph for drug embeddings. In a drug molecular graph $G = (a_i, E)$, each atom in the set of atoms $a_i$ is initialized with its feature vector $x_i$, and each bond $e_{ij}$ (between atoms $a_i$ and $a_j$) in the set of chemical bonds $E$ is represented by its feature vector $e_{ij}$.

In the message passing process, the MPNN updates the representation of each atom by aggregating its neighboring atoms. At each iteration $l - 1$, the message $me_i^{(l)}$ of atom $a_i$ is computed by aggregating its neighbors $\mathcal{N}(i)$ as follows:

$$me_i^{(l)} = \sum_{j \in \mathcal{N}(i)} f_m(h_i^{(l-1)}, h_j^{(l-1)}, e_{ij}) \tag{13}$$

where $f_m(\cdot)$ is a message function, and $h_i^{(l-1)}$ is the hidden state of atom $a_i$ at iteration $l - 1$. And then, the hidden state is updated with Eq. 14:

$$h_i^{(l)} = f_u(h_i^{(l-1)}, me_i^{(l)}) \tag{14}$$

where $f_u(\cdot)$ is the update function. With $L$ iterations of message passing, we obtain the final hidden states $h_i^{(L)}$ for all atoms.

These hidden states are then aggregated to obtain a fixed-size medication molecular feature vector $y_m$ for the one drug molecule. This readout operation can be performed with a pooling operation, which is shown as follows:

$$y_m = Pooling(\{h_i^{(L)} \mid i = 0, ..., n\}) \tag{15}$$

where $n$ is the number of atoms in the drug. We performed the aforementioned MPNN operation on all drug molecules, gathered the resulting feature vectors, and organized them into a drug molecule feature matrix $M_{mol} \in \mathbb{R}^{|M| \times dim}$.

*3.4.2 Score on Patient-medication Matching.* As the longitudinal patient representation $q_t$ input as a query, the medication representation module aims to identify the most relevant drugs from the drug molecular feature matrix $M_{mol}$. The matching score between the patient and each drug is calculated by taking the dot product of the MPNN-generated drug representations $M_{mol}$ and the longitudinal patient representation $q_t$, which is shown in Eq. 19:

$$m_{score}^{(t)} = \sigma(M_{mol} \cdot q_t) \tag{16}$$

For all drugs, each row of $M_{mol}$ is combined with $q_t$ and processed with an activation function $\sigma$. Here, each element in $m_{score}^{(t)}$ represents the matching score for a particular drug. To refine the process, the scores further flow into a feed-forward neural network, followed by layer normalization (LN) and a residual connection as shown in Eq. 20:

$$m_{internal}^{(t)} = LN(m_{score}^{(t)} + f(m_{score}^{(t)})) \tag{17}$$

where $f(\cdot)$ is the transform function implemented with a single hidden layer fully connected neural network.

*3.4.3 Functional-view of Drug Molecules.* Besides the internal-view of drug molecular features gained, we capture drug-drug interaction (DDI) information in a functional-view of drug molecules. Briefly, we leverage the graph attention network (GAT)[13] to effectively combine internal molecular features with drug functional molecules, ensuring comprehensive and accurate drug representations.

To capture the co-occurrence of drugs within the EHRs, we employ an adjacency matrix $A_{ehr}$ that reflects the presence of drugs within the same visit. Moreover, another adjacency matrix $A_{ddi}$ is also utilized to represent known DDIs, with the initial feature vectors $h_{m_i}$ for each drug node obtained from the molecular features vector $M_{mol}$. Therefore, the EHR graph $G_{ehr} = (A_{ehr}, M_{mol})$ and the DDI graph $G_{ddi} = (A_{ddi}, M_{mol})$ are available.

Subsequently, two GATs are employed to incorporate EHR information into DDI one, based on the dynamic weighting of interactions according to their relevance. In this way, the significant interactions are prioritized and medication representations are refined accordingly.

Taking the DDI graph as an example. With an attention mechanism equipped, a GAT is capable to compute the importance of each DDI. For each medication node $m_i$, the attention coefficient $\alpha_{ij}$ between nodes $m_i$ and $m_j$ is calculated as follows:

$$\alpha_{ij} = \frac{\exp(Leaky\,Re\,LU(a^T[Wh_{m_i}\|Wh_{m_j}]))}{\sum\limits_{k \in \mathcal{N}(i)} \exp(Leaky\,Re\,LU(a^T[Wh_{m_i}\|Wh_{m_k}]))} \tag{18}$$

where $W$ is the weight matrix, $a^T$ is the transpose of the weight vector of a single layer feed-forward neural network, and $\|$ denotes concatenation. Here, the LeakyReLU activation function is applied to introduce non-linearity.

As the attention coefficients are calculated, the updated feature vector for each drug node $m_i$ is obtained by aggregating its neighbors, and weighted summing of the attention coefficients as follows:

$$h_{m_i}^{'} = \sigma \left( \sum_{j \in \mathcal{N}(i)} \alpha_{ij} Wh_{m_j} \right) \tag{19}$$

where $\sigma$ denotes an activation function. As a result, the drug interaction representation $Z_{ddi} \in \mathbb{R}^{|M| \times dim}$ is generated by integrating the interaction information into the molecular feature vectors. In the same manner, the drug co-occurrence graph attention network is utilized to obtain the drug co-occurrence representation $Z_{ehr} \in \mathbb{R}^{|M| \times \dim}$.

Finally, a drug representation $m_{ed}$ is generated by integrating internal-view medication molecule structure with functional-view interactions in medication molecules, which is formulized as follows:

$$m_{ed} = Z_{ehr} + WZ_{ddi} \tag{20}$$

Additionally, the attention $\alpha_t$ can be computed based on the representation $m_{ed}$ and the longitudinal patient presentation $q_t$, which

**Table 1: Performance Comparison of Different Methods on MIMIC-III**

| Model | Jaccard | F1 Score | PRAUC | DDI Rate | # of Drugs |
|---|---|---|---|---|---|
| LR | 0.4949 ± 1.7e-05 | 0.6518 ± 1.621e-05 | 0.7559 ± 1.34e-05 | 0.06799 ± 9.68e-07 | 16.55 ± 0.0325 |
| ECC | 0.4807 ± 4.932e-06 | 0.6368 ± 4.76e-06 | 0.7560 ± 4.47e-06 | 0.07913 ± 7.77e-07 | **15.81 ± 0.0399** |
| RETAIN | 0.4834 ± 2.797e-06 | 0.6448 ± 2.17e-06 | 0.7598 ± 3.23e-07 | 0.08409 ± 2.62e-06 | 18.39 ± 0.2060 |
| LEAP | 0.4492 ± 3.912e-06 | 0.6113 ± 3.71e-06 | 0.6524 ± 3.94e-06 | 0.06789 ± 1.01e-06 | 18.82 ± 0.0025 |
| GAMENet | 0.5063 ± 1.191e-06 | 0.6624 ± 3.94e-07 | 0.7647 ± 7.05e-07 | 0.08329 ± 2.44e-06 | 25.94 ± 0.0338 |
| MICRON | 0.5174 ± 1.597e-06 | 0.6721 ± 1.26e-06 | 0.7739 ± 1.1e-07 | 0.06210 ± 5.11e-07 | 17.92 ± 0.0175 |
| SafeDrug | 0.5161 ± 2.655e-06 | 0.6720 ± 2.48e-06 | 0.7671 ± 6.48e-07 | 0.06165 ± 6.88e-07 | 19.6 ± 0.0983 |
| COGNet | 0.5141 ± 7.879e-06 | 0.6677 ± 8.39e-06 | 0.7413 ± 8.02e-05 | 0.08317 ± 6.46e-06 | 25.06 ± 0.1765 |
| **DG-MRM** | **0.5228 ± 1.182e-06** | **0.6778 ± 8.42e-07** | **0.7744 ± 3.54e-07** | **0.06153 ± 2.19e-07** | 19.29 ± 0.0903 |

is shown as follows:

$$\alpha_t = \sigma((m_{\text{ed}})^T \cdot q_t) \tag{21}$$

where $(m_{\text{ed}})^T$ is the transpose of $m_{\text{ed}}$. Therefore, the matching score of functional-view as follow:

$$m_{function}^{(t)} = \alpha_t \cdot m_{ed} \tag{22}$$

### 3.5 Medication Recommendation

The final output vector for medication recommendation is calculated as follows:

$$\hat{m}_t = f(q_t, m_{internal}^{(t)}, m_{function}^{(t)}) \tag{23}$$

where $f(\cdot)$ is the transform function implemented with a single hidden layer fully connected neural network.

## 4 COMPARATIVE EXPERIMENT

We compare the performance of various methods on the MIMIC-III dataset. The above results are derived from 10 rounds of bootstrap sampling, where each indicator was averaged and its variance calculated. Our proposed DG-MRM model outperforms all baselines in Jaccard, F1 score, and PRAUC, while maintaining a competitive DDI rate and fast convergence rates. Due to space limitation, we push other details to Appendix.

Firstly, instance-based models such as LR and ECC show lower performance due to their focus on current visit data while ignoring the historical information. Temporal models like RETAIN perform better by incorporating patients' visit history, though RETAIN's higher DDI rate is notable due to larger medication combinations.

Secondly, longitudinal approaches such as GAMENet, MICRON, and SafeDrug improve performance by integrating additional information. Specifically, GAMENet uses historical combinations and graph information, MICRON retains unchanged medications with a recurrent method, and SafeDrug incorporates molecular structures to reduce DDI rates. COGNet utilizes the similarity of historical visits for medication recommendation, achieving good results. However, the inherent DDI rate in historical prescriptions (around 0.08) leads to a higher DDI rate in COGNet's results, and it tends to recommend more medications. Nevertheless, these models still struggle to balance accuracy and safety.

Finally, Our DG-MRM model achieves the best scores in Jaccard, F1, PRAUC and DDI rate, demonstrating its superior accuracy and safety in medication recommendations. Notably, DG-MRM maintains a lower DDI rate compared to most baselines, highlighting

its effectiveness in minimizing adverse DDIs. This success is due to the dynamic graph construction and multi-perspective integration in DG-MRM, which results from its accurate capturing the patients' structural relationships in EHR data. Additionally, the architectural advantage of DG-MRM lies in its ability to effectively capture semantic features and structural relationships in patients' visit sequences, providing a robust solution for the medication recommendation task. Overall, DG-MRM excels in both accuracy and safety in comparison experiments.

## 5 CONCLUSION

In this paper, we propose DG-MRM, a novel drug recommendation model that utilizes dynamic graph construction and multi-perspective medication information to enhance the safety and efficacy of medication recommendations. Our model integrates temporal evolution and structural relationships within patients' EHR data, alongside controlling DDI information. Experimental results on the MIMIC-III dataset demonstrate that DG-MRM outperforms the state-of-the-art baseline models, achieving higher scores in Jaccard, F1, and PRAUC metrics while maintaining a lower DDI rate. These improvements highlight the model's capability to provide accuracy and safety in medication recommendation tasks.

In the future research, we will further investigate the structural relationships in EHRs, and explore personalized medication recommendation based on the knowledge supporting from medical knowledge bases. With the powerful learning ability of Large Models, we direct our energies to personalized precise medication recommendation with controllable DDIs.

## ACKNOWLEDGMENTS

This work is supported by the Natural Science Foundation of Shandong Province, China (No.ZR2021MF118, No.ZR2020LZH008, No.ZR2022LZH003), the Key R&D Program of Shandong Province, China (No.2021CXGC010506, No.2021SFGC0104), and the National Natural Science Foundation of China (No.62101311, No.62072290), Youth Science Foundation Project of Shandong Province (No.ZR2022QF022), Postgraduate Quality Education and Teaching Resources Project of Shandong Province (No.SDYKC2022053, No.SDYAL2022060), and Jinan "20 new colleges and universities" Funded Project (No.202228110).

Guanlin Liu[1], Zihao Liu[1], Xiaomei Yu[1,2,*], Xue Li[1], Xiangwei Zheng[1,2], and Xingxu Fan[1]

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
