# OpenReview forum: "Dynamic Graph and Multi-view Medication Information for Recommending Medication Combination"
_KDD.org/2024/Workshop/AIDSH — KDD-AIDSH 2024 Poster_

### Official Review · Reviewer_4tQM · 2024-06-16
**Well-written paper, extensive experiments and effective method for Safe Medication Recommendation**

**Rating:** 9
**Confidence:** 4

**Review:**

Summary Of Strengths:
- Simple, effective way for recommending safe and accurate medication combination
- The paper extensively evaluates the proposed method compared to current methods and provides clear ablation to show the effectiveness of each proposed component.
- Well-written and easy to follow

Summary Of Weaknesses:
- It's better to provide interpretability and cases to show the actual medication combination for a patient and how it dynamically evolves among the patient's multiple visits
- Minor typos should be fixed, such as Chinese words in Figure 1 (\#Line 430)

---

### Official Review · Reviewer_6jrP · 2024-06-20

**Rating:** 8
**Confidence:** 4

**Review:**

### Brief Summary

The paper presents DG-MRM, a novel medication recommendation model that addresses the challenges of accurately and safely recommending medication combinations. The model constructs dynamic graphs from multi-source Electronic Health Records (EHR) data and integrates multi-view medication information, including molecular structures and drug-drug interaction data. DG-MRM leverages Recurrent Neural Networks (RNNs) to capture temporal dynamics and employs graph neural networks to investigate the relationships between different medical events. The model aims to provide comprehensive patient representations and generate safe medication combinations by considering both the molecular structure of medications and the interactions between them.

### Strengths
- Innovative Approach: The integration of dynamic graph neural networks with multi-view medication information is a novel approach that captures the complexity of patient data and medication interactions.
- Comprehensive Representation: The model's ability to generate comprehensive patient representations by considering both temporal and structural relationships in EHR data is a significant strength.
- Safety and Efficacy: DG-MRM's focus on balancing the safety and efficacy of medication recommendations by incorporating drug-drug interaction information is crucial for clinical applications.
- Empirical Validation: The model has been rigorously evaluated on a benchmark dataset, demonstrating improvements over existing methods in terms of efficacy and safety.

### Weaknesses
- Generalizability: The paper does not provide enough information on how well the model generalizes to different populations or healthcare settings. I would like to suggest considering more ICU datasets in addition to MIMIC.
- Dependence on Data Quality: The model's effectiveness is likely dependent on the quality and completeness of the EHR data, which can vary significantly across institutions. Would this approach with dynamic graph and multi-view representations be robust to some data missing or noises?

---

### Decision · Program_Chairs · 2024-06-28

Accept (Poster)